# *TET1* Knockdown Inhibits *Porphyromonas gingivalis* LPS/IFN-γ-Induced M1 Macrophage Polarization through the NF-κB Pathway in THP-1 Cells

**DOI:** 10.3390/ijms20082023

**Published:** 2019-04-24

**Authors:** Yanlan Huang, Cheng Tian, Qimeng Li, Qiong Xu

**Affiliations:** Guanghua School of Stomatology & Guangdong Provincial Key Laboratory of Stomatology, Sun Yat-sen University, 56 Ling Yuan Xi Road, Guangzhou 510055, China; huangylan5@mail2.sysu.edu.cn (Y.H.); tianch5@mail2.sysu.edn.cn (C.T.); liqimeng22@outlook.com (Q.L.)

**Keywords:** M1 macrophage polarization, *Porphyromonas gingivalis*, LPS/IFN-γ, TET1, NF-κB

## Abstract

Tet-eleven translocation 1 (TET1) is a dioxygenase that plays an important role in decreasing the abundance of DNA methylation and changing the expression levels of specific genes related to inflammation. *Porphyromonas gingivalis* (Pg.) lipopolysaccharide (LPS) can induce periodontal diseases that present with severe bone loss and collagen fiber destruction accompanied by a high number of M1 macrophages. M1-polarized macrophages are pivotal immune cells that promote the progression of the periodontal inflammatory response, but the function of TET1 during M1 macrophage activation is still unknown. Our results showed that the mRNA and protein expression levels of TET1 decreased in THP-1 cells during M1 macrophage differentiation. *TET1* knockdown resulted in a significant decrease in the production of proinflammatory markers such as IL-6, TNF-α, CCL2, and HLA-DR in Pg. LPS/IFN-γ- and *Escherichia coli* (*E. coli*) LPS/IFN-γ-induced M1 macrophages. Mechanistically, *TET1* knockdown downregulated the activity of the NF-κB signaling pathway. After treatment with the NF-κB inhibitor BAY 11-7082, M1 marker expression showed no significant difference between the *TET1* knockdown group and the control group. Taken together, these results suggest that *TET1* depletion inhibited Pg. LPS/IFN-γ-induced M1 macrophage polarization through the NF-κB pathway in THP-1 cells.

## 1. Introduction

Macrophage activation, which has an indispensable role in maintaining immune system homeostasis, is involved in the immunopathology of multiple inflammatory diseases such as diabetes, atherosclerosis, gingivitis, and periodontitis [1,2,3,4]. According to differences in pathogenic infections or stimulation factors, heterogeneous macrophages can be classified as classically activated macrophages (M1) and alternatively activated macrophages (M2) [5]. While the M1 program typically contributes to powerful antigen presentation and high microbicidal activity, the M2 program generally produces macrophages that participate in anti-inflammatory functions such as tissue repair and tumor invasion [6]. In the presence of Th1 factors such as lipopolysaccharide (LPS) and interferon-γ (IFN-γ), M1 macrophages produce proinflammatory mediators such as tumor necrosis factor-α (TNF-α), interleukin-6 (IL-6), class II major histocompatibility complex (HLA-DR), and C-C motif chemokine ligand 2 (CCL2) [7]. The secretion of M1 macrophage proinflammatory molecules is modulated by the activation of nuclear factor κ-light-chain-enhancer of activated B cells (NF-κB), activator protein 1 (AP-1), and interferon-regulatory factors (IRFs) [8,9,10].

Periodontal diseases, which include gingivitis and periodontitis, are progressive inflammatory diseases caused by bacterial infection accompanied by the host immune response, periodontal tissue destruction, and alveolar bone loss [11]. The gram-negative bacteria *Porphyromonas gingivalis* (Pg.) is one of the most important periodontal pathogens that is frequently observed in the subgingival biofilm [12,13]. Pg. infection increases the necroptosis rate among defense cells and promotes severe bone destruction in the periodontal tissue [14]. Murine teeth injected with Pg. LPS show histological lesions that many vacuolated macrophages infiltrate and alveolar bone resorption around the periodontal ligament lesion [15]. A higher proportion of M1 macrophages correlates positively with periodontitis [16]. The M1-related cytokines IL-1β and IL-6, which are secreted by macrophages, induce the expression of matrix metalloproteinases (MMPs) in human gingival fibroblasts (HGFs). MMPs then cause the destruction of gingival collagen fibers in inflamed periodontal tissue under high-glucose conditions [17]. These findings demonstrate that M1 macrophages might play an important role in the occurrence and development of periodontitis.

DNA methylation is an important epigenetic modification that serves as a critical switch for gene expression and widely impacts cellular physiological functions. DNA methylation is dynamically balanced by methylation and demethylation. While DNA methyltransferases (DNMTs) serve as methylation writers and maintainers to generate 5-methylcytosine (5mC), ten-eleven translocation proteins (TETs) act as methylation erasers to oxidize 5mC into 5-hydroxymethylcytosine (5hmC) [18,19]. TETs are Fe^2+^- and α-ketoglutarate (α-KG)-dependent dioxygenases that are classified as TET1, TET2, and TET3. Increasing evidence has demonstrated that TETs play important roles in tumor progression and embryonic development. Recently, TETs have gained attention in the inflammatory process. Upon *Escherichia coli* (*E. coli*) LPS stimulation, TET1 downregulates *pro-IL-1β* and *IL-1β* transcription in the THP-1 cell line [20]. In regulatory T cells (Tregs), transforming growth factor-β (TGF-β)-activated Smad3 and interleukin-2 (IL-2)-activated signal transducers and activators of transcription 5 (Stat5) induce Tet1 and Tet2 binding to the fork-head-family transcription factor (*Foxp3*) promoter. Tet1 and Tet2 then enhanced the *Foxp3* promoter hypomethylation, which promoted Treg differentiation to further regulate the autoimmune response [21]. TET2 participates in the LPS-induced inflammatory response in human dental pulp cells by epigenetically regulating the transcription of the NF-κB signaling pathway transduction molecule MyD88 [22]. Thus, TET proteins can impact the expression of particular genes and change the process of inflammation by different mechanisms. Our previous study found that the level of TET1 markedly decreased while the levels of TET2 and TET3 showed no significant changes during Pg. LPS/IFN-γ-induced M1 macrophage polarization in THP-1 cells. However, whether TET1 is involved in the regulation of macrophage activation during the periodontal inflammatory process is still unknown.

In this study, we examined the expression of TET1 and the consequences of *TET1* knockdown during M1 macrophage polarization in THP-1 cells. Furthermore, we investigated whether TET1 inhibits the NF-κB signaling pathway during M1 macrophage activation. As a result, we found that TET1 modulates M1 macrophage differentiation by the NF-κB pathway.

## 2. Results

### 2.1. Pg. LPS/IFN-γ Polarized M0 Macrophages into M1 Proinflammatory Macrophages

To verify that monocytes differentiated into naive macrophages, the expression of the human cluster of differentiation 68 (CD68) was detected by fluorescence-activated cell sorting analysis (FACS) [23]. After PMA treatment for 24 h, the expression of CD68 was significantly increased (Figure 1A). Morphological evidence showed that THP-1 cells were round and small, while M0 macrophages were clearly larger and contained many granules when viewed under an inverted microscope (Figure 1B). Thus, the THP-1 cells were differentiated into macrophages (M0).

To transform M0 macrophages into M1 proinflammatory macrophages, the cells were treated with LPS/IFN-γ, as previously described [24,25]. For cell viability, no significant difference was shown for 0, 1, and 10 ng/mL IFN-γ at 24 and 48 h. A higher concentration of IFN-γ treatments suppressed the cell proliferation rates. No significant difference was shown for 0-1000 ng/mL Pg. LPS and 10 ng/mL IFN-γ within 72 h (Figure 2B). Cell viability was inhibited in the groups above 0.1 ng/mL *E. coli* LPS and 10 ng/mL IFN-γ at 48 h and 72 h (Figure 2C). The treatments of 100 ng/mL Pg. LPS/10 ng/mL IFN-γ and 0.1 ng/mL *E. coli* LPS/10 ng/mL IFN-γ were chosen to establish the M1 macrophage model.

To verify the efficiency of M1 macrophage activation, the expression levels of classic M1 markers, such as *IL-6*, *TNF-α*, *CCR7*, *CCL2*, and HLA-DR, were assessed by a real-time quantitative polymerase chain reaction (RT-qPCR) and FACS. Following Pg. LPS/IFN-γ or *E. coli* LPS/IFN-γ stimulation, the expression levels of M1 markers significantly increased (Figure 2D–G). Both Pg. LPS/IFN-γ and *E. coli* LPS/IFN-γ promoted round M0 macrophages to develop a flat morphology with spindle pseudopodia that could be viewed with an inverted microscope, which further confirms the LPS-IFN-γ-induced M1 macrophage polarization (Figure 1B).

### 2.2. TET1 Expression Decreased in LPS/IFN-γ-Induced M1 Macrophages

To study the TET1 expression pattern during the generation of M1 macrophages, we detected the TET1 mRNA and protein levels in Pg. LPS/IFN-γ or *E. coli* LPS/IFN-γ-induced macrophages. The mRNA expression of *TET1* was repressed after LPS/IFN-γ treatment at 6 h, 12 h, and 24 h. The suppression of *TET1* mRNA expression was more clear with *E. coli* LPS/IFN-γ treatment than with Pg. LPS/IFN-γ stimulation (Figure 3A,C). The TET1 protein level also decreased in Pg. LPS/IFN-γ- and *E. coli* LPS/IFN-γ-treated cells (Figure 3B,D).

### 2.3. TET1 Knockdown Inhibited the Expression of Proinflammatory Cytokines and Chemokines during LPS/IFN-γ-Induced M1 Macrophage Activation

To evaluate the function of TET1 in M1 macrophage activation, *TET1* siRNAs or a control siRNA were transfected into M0 cells. Compared to the control group, the #3 *TET1* siRNA group exhibited a decrease in TET1 expression of up to 70% to 80% at both the mRNA and protein levels (Figure 4A–C). Therefore, the #3 *TET1* siRNA sequence resulted in the best knockdown effect and was further utilized in the following experiments.

To explore the effect of *TET1* knockdown on M1 macrophage activation, cells were treated with Pg. LPS/IFN-γ or *E. coli* LPS/IFN-γ for 6, 12, and 24 h, and M1 markers were analyzed after *TET1* knockdown. Following LPS/IFN-γ treatment, the mRNA levels of *IL-6*, *TNF-α*, and *CCL2* showed different degrees of decrease in the *TET1* knockdown group (Figure 5A). The IL-6 protein level significantly decreased in the *TET1* knockdown group compared with the control group, and TNF-α expression was decreased at 24 h after LPS/IFN-γ stimulation (Figure 5B). *TET1* knockdown slightly inhibited the induction of HLA-DR protein in *E. coli* LPS/IFN-γ-induced macrophages, but did not affect HLA-DR expression in Pg. LPS/IFN-γ-induced M1 macrophages at 24 h (Figure 5C).

### 2.4. TET1 Knockdown Attenuated M1 Macrophage Polarization through the NF-κB Signaling Pathway

The NF-κB signaling pathway plays an indispensable role in M1 macrophage activation. To investigate the role of TET1 in the regulation of the NF-κB signaling pathway during M1 macrophage activation, the expression of the phosphorylated forms of the key factors IKKα/β, IκBα, and p65 was examined by Western blot analysis. The results revealed that *TET1* knockdown decreased the levels of phosphorylated IKKα/β, IκBα, and p65 in LPS/IFN-γ-induced M1 macrophages at the indicated time points (Figure 6A–D). IκB kinase (IKK) consists of two subunits including IKKα and IKKβ. Its activation facilitates the phosphorylation, ubiquitination, and proteasomal degradation of IκBα [26]. IkBα is a molecular inhibitor of NF-κB and its phosphorylation drives its ubiquitination and proteasome degradation [27]. As an effector member of NF-κB pathway, p65 can be liberated from IκBα and then phosphorylates and translocates into the nucleus, which indicates the activation of NF-κB signaling [28]. The levels of phosphorylated IKKα/β, IκBα, and p65 decreased in the *TET1* knockdown group during M1 macrophage activation, which indicates that *TET1* depletion inhibited the NF-kB signaling pathway.

To further evaluate whether TET1 regulated proinflammatory cytokine expression through the NF-κB signaling pathway during LPS/IFN-γ-induced M1 macrophage activation, the NF-κB inhibitor BAY11-7082 was used to block this signaling pathway. Then, the expression levels of proinflammatory M1 markers were assessed. The results showed that the downregulation of *IL-6*, *TNF-α*, and *CCL2* by *TET1* depletion showed no difference in LPS/IFN-γ-induced M1 macrophages after the NF-κB signaling pathway was blocked (Figure 6E).

## 3. Discussion

Periodontitis is an inflammatory disease caused by infection with periodontal pathogens such as Pg., which results in tissue destruction in the periodontal tissue accompanied by extensive macrophage infiltration [29,30,31]. Macrophages are heterogeneous and plastic cells that can be polarized into different phenotypes and exhibit various functions in response to surrounding stimuli [32]. The cell walls of gram-negative bacteria contain LPS, which can trigger the inflammatory response and generate M1 macrophages with or without IFN-γ [33,34]. In response to Pg. infection, M1 macrophages widely infiltrate and cause severe bone destruction in the periodontium [35]. An interesting phenomenon was discovered, and it showed that M1 macrophage numbers are significantly increased and there is a small change in the M2 phenotype in periodontitis in nonhuman primates [36]. Pg. LPS can induce the NF-κB signaling pathway activation to generate M1-related cytokines such as IL-6, TNF-α, IL-1β, and nitric oxide (NO) in macrophages during the progression of periodontitis [30,37]. In a mouse periodontitis model established by exposure to Pg., human β-defensin 3 (hBD3) reduces M1 macrophage polarization by inhibiting the phosphorylation of the NF-κB p65 subunit, which results in less alveolar bone loss [38]. To verify the function of Pg. LPS/IFN-γ in M1 macrophage differentiation, we examined the expression levels of M1 markers such as IL-6, TNF-α, CCR7, CCL2, and HLA-DR in THP-1-derived macrophages. Since *E. coli* LPS/IFN-γ treatment is a classic protocol to induce M1 macrophages, this treatment was chosen as a positive control for the induction of M1 macrophages in the present study. We demonstrated that the mRNA and protein levels of the M1 markers were reinforced after Pg. LPS/IFN-γ treatment, which suggests that Pg. LPS/IFN-γ induced M1 polarization in macrophages. Compared to *E. coli* LPS, which has well-defined activity, Pg. LPS showed less biological reactivity.

As one of the most important epigenetic modifications, DNA methylation is considered to be an important factor for the inflammatory response and immune cell differentiation [39]. A previous study showed that DNMTs promoted the expression of genes involved in inflammation, such as IL-6, TNF-α, IL-1β, NO, and CCL2, in mouse macrophages via the methylation of CpG-enriched sites in the liver X receptor α (*LXRα*) and peroxisome proliferator-activated receptor γ1 (*PPARγ1*) promoters. LXRα and PPARγ1 could inhibit the ability of NF-κB to drive inflammatory gene expression [40]. The combination of the DNMT inhibitor 5-Aza-2-deoxycytidine (Aza) and the histone deacetylase (HDAC) inhibitor Trichostatin A (TSA) decreased the number of LPS-induced M1 macrophages and increased the number of M2 macrophages by inhibiting HuR and activating the STAT3-Bcl2 pathways [41]. Compared to DNA methylation mediated by DNMTs, the role of DNA demethylation during macrophage activation and inflammation has been investigated in only a few studies. To evaluate the role of the DNA demethylase TET1 in M1 macrophage activation, we detected the expression level of TET1 in THP-1 cells after Pg. LPS/IFN-γ treatment. The trend of TET1 expression level was contrary to that of the M1 markers IL-6, TNF-α, CCR7, and CCL2 in THP-1-derived macrophages induced with Pg. LPS/IFN-γ treatment. This result indicated that TET1 might participate in the regulation of Pg. LPS/IFN-γ-stimulated macrophage M1 activation.

Several lines of evidence support the conclusion that TET1 is relevant for activating immune cells [42,43]. High TET1 expression is related to lower infiltration by immune cells, such as T and B cells, in basal-like breast cancer [44]. In an inflammatory rat burn injury model, elevated Tet1 expression helps to demethylate the M1-related gene *iNOS*, which results in the upregulation of expression and the inflammatory response [45]. Melatonin alleviates inflammation and increases the ratio of M2 to M1 macrophages by transporting exosomes from adipose cells. Macrophages absorb exosomes enriched in α-KG and promote TET-mediated DNA demethylation in adipose tissue [46]. To explore the effect of TET1 on the inflammatory response in Pg. LPS/IFN-γ-induced M1 macrophages, we silenced *TET1* expression in THP-1 cells and detected the mRNA and protein levels of M1-related cytokines. The data showed that the expression levels of IL-6, TNF-α, and CCL2 decreased after *TET1* knockdown, which indicates that TET1 could positively regulate M1 macrophage polarization.

The NF-κB signaling pathway plays an important role in the inflammatory response and macrophage polarization. A growing body of evidence showed that the NF-κB signaling pathway can be regulated by DNA methylation [47,48]. In germinal center B cells, elevated DNMT1 expression causes hypomethylation of the promoter of the nuclear factor kappa B kinase subunit epsilon (*IKBKE*) inhibitor, which is one of the NF-κB pathway signaling components that induces IKB phosphorylation [47]. The DNA methyltransferase inhibitor Aza facilitates the nuclear translocation of NF-κB and promotes the TLR4-NF-κB signaling pathway [49]. Aza treatment can promote the activation of IKKa/β and IκBα and facilitates, which results in high cell survival and proliferation in gastric cancer cells [50]. To determine whether TET1 influences the activation of the NF-κB pathway during Pg. LPS/IFN-γ-induced macrophage M1 polarization, we detected the phosphorylation levels of several key signaling molecules in the NF-κB signaling pathway by Western blot analysis. The results showed that *TET1* depletion reduced the phosphorylation levels of IKKα/β, p65, and IκBα under LPS/IFN-γ treatment conditions. To further clarify the function of the NF-κB signaling pathway in M1 macrophage activation, an NF-κB inhibitor was used to block this signaling pathway. After treatment with the NF-κB inhibitor, the mRNA expression levels of the proinflammatory markers *IL-6*, *TNF-α*, and *CCL2* showed no difference between the *TET1*-deficient group and the control group. These data demonstrated that *TET1* depletion reduced M1 macrophage polarization by inhibiting the activation of the NF-κB signaling pathway.

In conclusion, this study indicated that TET1 expression decreased during Pg. LPS/IFN-γ-induced M1 polarization in THP-1 macrophages. Under Pg. LPS/IFN-γ stimulation conditions, *TET1* knockdown inhibited M1 macrophage polarization through the NF-κB signaling pathway. These findings may offer a new strategy to research macrophage polarization to regulate the inflammatory response.

## 4. Materials and Methods

### 4.1. Cell Culture

THP-1 cells from the American Type Culture Collection (ATCC, Manassas, VA, USA) were cultured in RPMI-1640 medium supplemented with 10% fetal bovine serum (FBS, Gibco, Carlsbad, CA, USA) at 37 °C in an atmosphere of 95% O_2_ and 5% CO_2_. To generate M0 macrophages (M0), THP-1 cells were treated with 100 ng/mL phorbol 12-myristate 13-acetate (PMA, AdipoGen, San Diego, CA, USA) for 24 h [51].

For the NF-κB signaling pathway inhibition, M0 macrophages were treated with 2 µl/mL BAY11-7082 (Beyotime, Haimen, China) for one hour before further experiments.

### 4.2. M1 Macrophage Polarization

To obtain M1 macrophages, M0 cells were incubated with 100 ng/mL Pg. LPS (InvivoGen, San Diego, CA, USA) and 10 ng/mL IFN-γ (Cyagen, Santa Clara, CA, USA) and collected at the indicated time. *E. coli* LPS (0.1 ng/mL, Sigma-Aldrich, St. Louis, MO, USA) and IFN-γ (10 ng/mL)-treated macrophages were used as a positive control.

### 4.3. Cell Counting Kit-8 (CCK8) Assay

A CCK8 assay tested the proliferation of cells (Dojindo, Kumamoto, Japan) according to the manufacturer’s instructions. After 24, 48, or 72 h of incubation, the supernatant was refreshed with RPMI-1640 medium containing CCK-8 for another 2 h. An automated microplate reader (Sunrise, Tecan, Hombrechtikon, Switzerland) was used to read the optical density (OD) at 450 nm.

### 4.4. TET1 Small Interfering RNA (siRNA) Transfection

When M0 macrophages reached 60% confluence, they were transfected with a *TET1* siRNA (100 nM) or negative control siRNA with Lipofectamine RNAi MAX (Invitrogen, Carlsbad, CA, USA), according to the manufacturer’s instructions. The siRNA sequences are listed in Table 1.

### 4.5. Real-Time Quantitative Polymerase Chain Reaction (qRT-PCR)

Cells were lysed with the TRIzol reagent (Invitrogen, Carlsbad, CA, USA) and RNA was reverse transcribed into complementary DNA (cDNA) by PrimeScript™ RT Master Mix (Takara, Tokyo, Japan) following the manufacturer’s instructions. Then, the cDNA was used as a template for PCR with a LightCycler 480 thermocycler and SYBR green I Master Mix (Roche, Basel, Switzerland). Primers were designed by using Primer Express v3.0 software (Life Technologies Corp., Carlsbad, CA, USA) and are shown in Table 2.

### 4.6. Western Blot Analysis

Total protein was extracted by using the RIPA buffer (Beyotime, Haimen, China) and measured by using a BCA protein assay kit (ComWin Biotech, Beijing, China). A total of 30 μg of protein was subjected to 10% sodium dodecyl sulfate-polyacrylamide gel electrophoresis and transferred to polyvinylidene fluoride membranes (Millipore, Billerica, MA, USA). The membranes were blocked with 5% bovine serum albumin for 1 h and incubated with the following primary antibodies against TET1 (GeneTex, Irvine, CA, USA), IKKα/β, phospho-IKKα/β, p65, phospho-p65, IκBα, phospho-IκBα, and VINCULIN (Cell Signaling Technology, CST, Danvers, MA, USA). After washing, the membranes were incubated with secondary antibodies (CST, Danvers, MA, USA) for 1 h. The proteins were visualized using an enhanced chemiluminescence system (Millipore, Billerica, MA, USA) and analyzed by ImageJ software (National Institutes of Health, Baltimore, MD, USA).

### 4.7. Enzyme-Linked Immunosorbent Assay (ELISA)

ELISA kits were used to test the IL-6 and TNF-α protein concentrations in culture supernatants (R&D Systems, Minneapolis, MN, USA) according to the manufacturer’s instructions. A microplate reader (Tecan, Hombrechtikon, Switzerland) was used to detect the optical density at 450 nm.

### 4.8. Fluorescence-Activated Cell Sorting Analysis (FACS)

Cells were washed and stained for 20 min at room temperature with monoclonal antibodies specific for CD68 and HLA-DR (eBioscience, Carlsbad, CA, USA). Then, the samples were analyzed by flow cytometry (Beckman, San Francisco, CA, USA) and FlowJo software.

### 4.9. Statistical Analysis

Each experiment was repeated at least three times. All data were statistically analyzed by using the SPSS 20.0 software package (SPSS Inc., Chicago, IL, USA). The differences between groups were evaluated by a Student’s *t*-test. *p* < 0.05 was considered statistically significant.

## Figures and Tables

**Figure 1 ijms-20-02023-f001:**
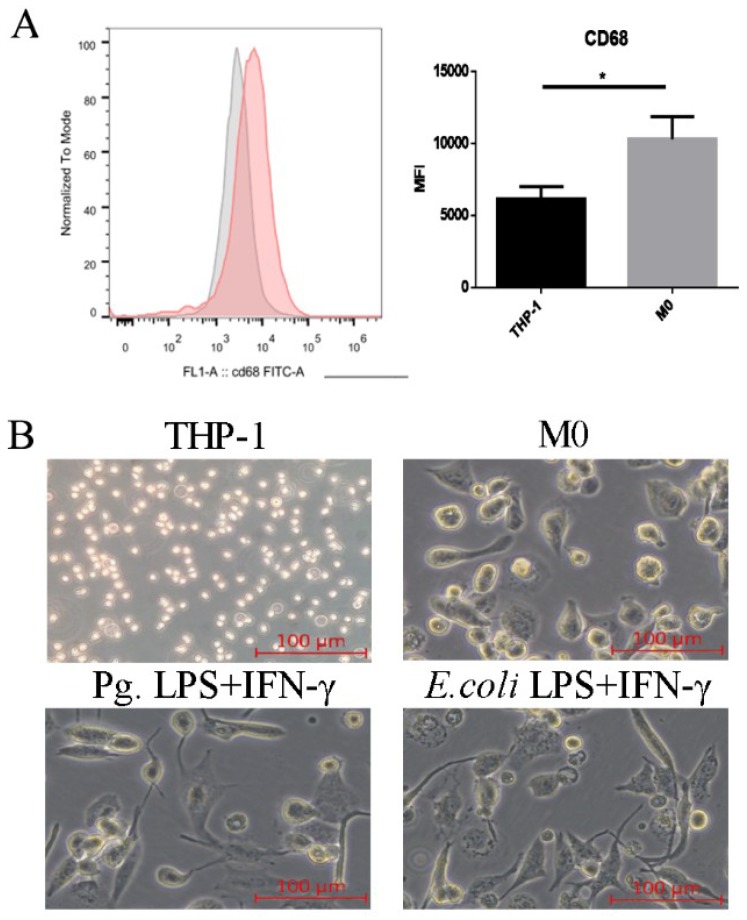
Differentiation of THP-1 cells to generate macrophages. (**A**) Flow cytometry analyzed the expression of CD68 by THP-1 cells and M0 macrophages. (**B**) The cells were treated with 1640 complete medium (THP-1 group), 100 ng/mL PMA (M0 group), and 100 ng/mL Pg. LPS+10 ng/mL IFN-γ or 0.1 ng/mL *E. coli* LPS+10 ng/mL IFN-γ for 24 h. Morphological changes were visualized by a phase contrast inverted microscope under 100 x magnification. All of the results represent the mean ± standard deviation of three independent experiments (*n* = 3). * *p* < 0.05 indicates a significant difference compared with the control.

**Figure 2 ijms-20-02023-f002:**
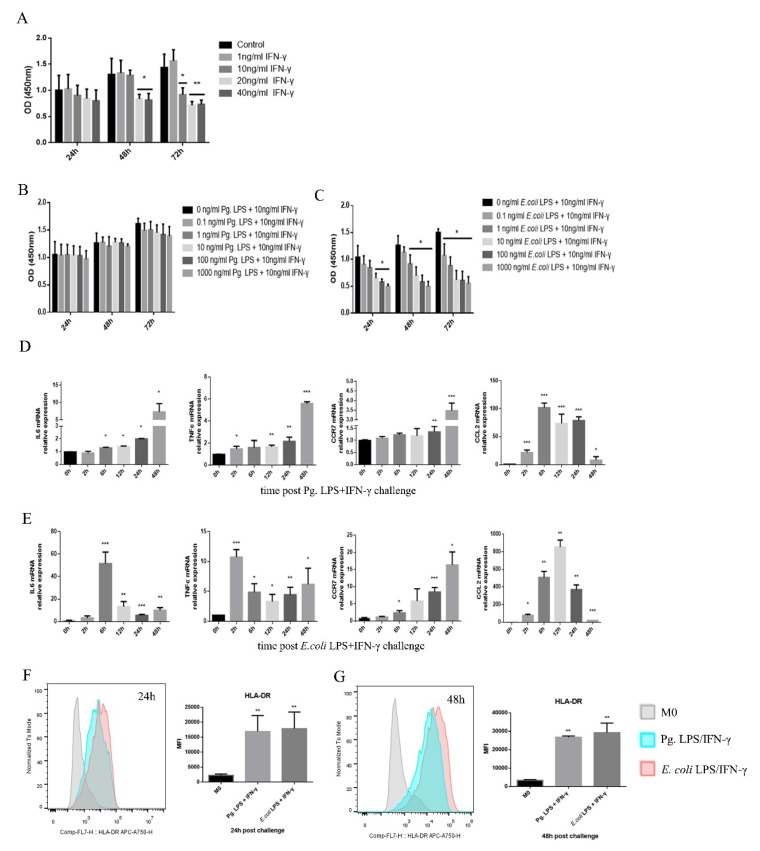
LPS/IFN-γ-induced M1 proinflammatory macrophage differentiation from M0 macrophages. (**A**–**C**) The effects of IFN-γ (**A**), Pg. LPS/IFN-γ, (**B**) and *E. coli* LPS/IFN-γ (**C**) on the proliferation of THP-1-derived macrophages were assessed. The cell growth of each group was measured by a CCK8 assay. (**D**,**E**) The mRNA expression of *IL-6*, *TNF-α*, *CCR7*, and *CCL2* was measured by qRT-PCR in cells after stimulation with Pg. LPS/IFN-γ (**D**) or *E. coli* LPS/IFN-γ (**E**) for 2, 6, 12, 24, and 48 h. *GAPDH* was used as an internal control. (**F**,**G**) The expression of HLA-DR in macrophages was analyzed by flow cytometry. M0 macrophages were used as a control. All of the results represent the mean ± standard deviation of three independent experiments (*n* = 3). * *p* < 0.05, ** *p* < 0.01, and *** *p* < 0.0001 represent significant differences compared with the control.

**Figure 3 ijms-20-02023-f003:**
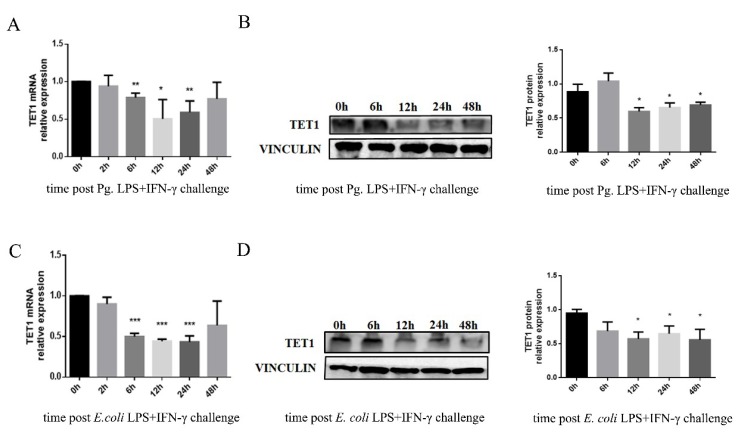
TET1 expression in M1 macrophages during LPS/IFN-γ stimulation. (**A**,**C**) The mRNA expression of *TET1* was measured by qRT-PCR in cells after stimulation with Pg. LPS/IFN-γ (**A**) or *E. coli* LPS/IFN-γ (**C**) for 2, 6, 12, 24, and 48 h. *GAPDH* was used as an internal control. (**B**,**D**) The protein expression of TET1 was measured by Western blot analysis after stimulation with Pg. LPS/IFN-γ (**B**) or *E. coli* LPS/IFN-γ (**D**) for 6, 12, 24, and 48 h. VINCULIN was used as an internal control. All of the results represent the mean ± standard deviation of three independent experiments (*n* = 3). * *p* < 0.05, ** *p* < 0.01, and *** *p* < 0.0001 represent significant differences compared with the control.

**Figure 4 ijms-20-02023-f004:**
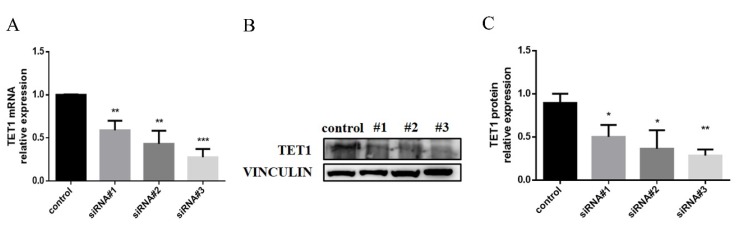
The effect of the siRNA-mediated depletion of TET1 expression in THP-1 cells. The mRNA (**A**) and protein (**B**,**C**) expression of TET1 was measured by qRT-PCR and Western blot analysis in THP-1 induced macrophages following siRNA treatment. VINCULIN was used as an internal control. All of the results represent the mean ± standard deviation of three independent experiments (*n* = 3). * *p* < 0.05, ** *p* < 0.01, and *** *p* < 0.0001 represent significant differences compared with the control.

**Figure 5 ijms-20-02023-f005:**
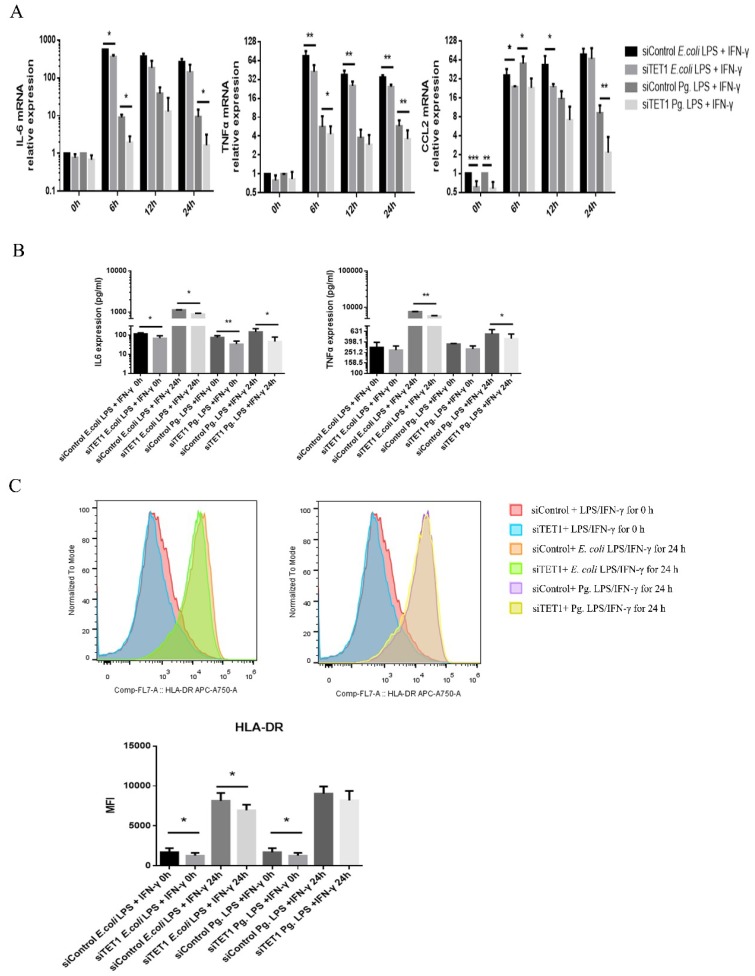
The effect of *TET1* knockdown on proinflammatory markers expressed during LPS/IFN-γ-induced M1 macrophage activation. (**A**) qRT-PCR assessed the mRNA expression levels of *IL-6*, *TNF-α*, and *CCL2* in LPS/IFN-γ-induced M1 macrophages. (**B**) The enzyme-linked immunosorbent assay (ELISA) was used to assess the protein expression of IL-6 and TNF-α in LPS/IFN-γ-induced M1 macrophages. (C) The protein expression of HLA-DR was analyzed by flow cytometry. All of the results represent the mean ± standard deviation of three independent experiments (*n* = 3). * *p* < 0.05 and ** *p* < 0.01 represent significant differences when compared with the control.

**Figure 6 ijms-20-02023-f006:**
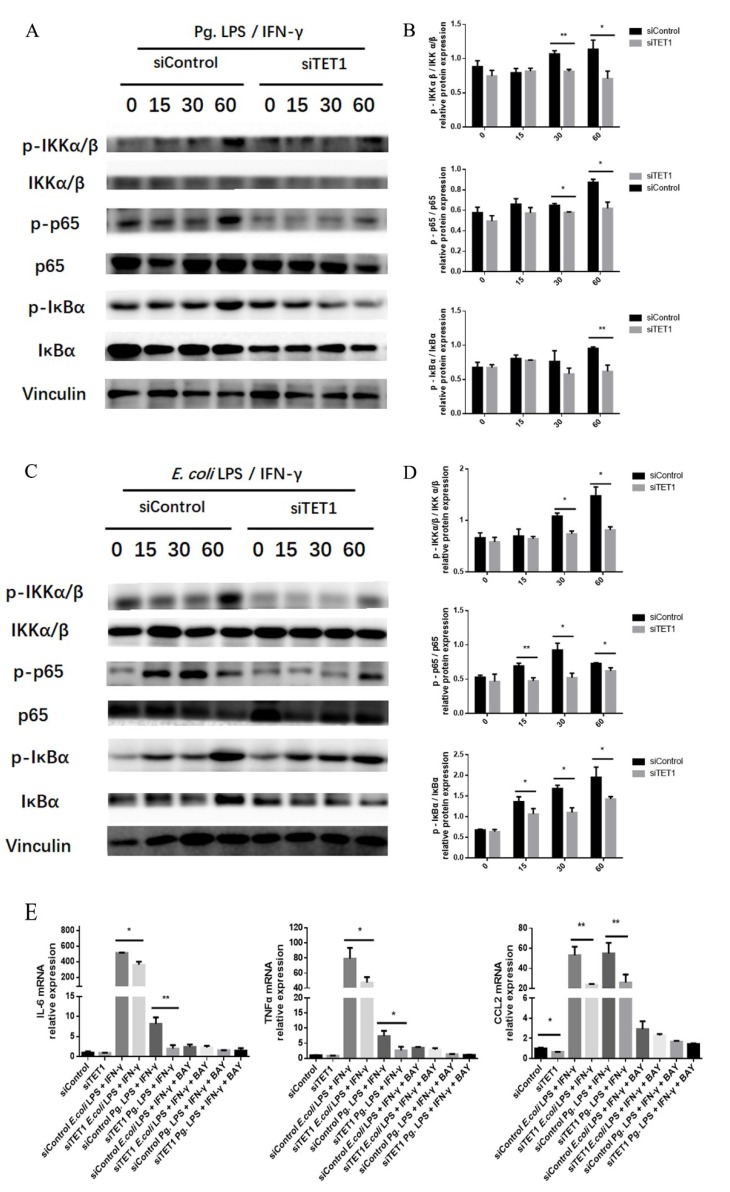
The effect of *TET1* depletion on the NF-κB signaling pathway during LPS/IFN-γ-induced M1 macrophage polarization. (**A**,**C**) The levels of the phosphorylated forms of key NF-κB signaling pathway proteins were measured by Western blot analysis (**A**: Pg. LPS/IFN-γ. **C**: *E. coli* LPS/IFN-γ). VINCULIN was used as an internal control. (**B**,**D**) The relative quantitative analysis of the phosphorylation levels of IKKα/β, p65, and IκBα was performed by Western blot analysis (**B**: Pg. LPS/IFN-γ. **D**: *E. coli* LPS/IFN-γ). (**E**) The mRNA expression levels of *IL-6*, *TNF-α*, and *CCL2* were assessed by qRT-PCR with/without pretreatment with the inhibitor BAY11-7082. *GAPDH* was used as an internal control. All of the results represent the mean ± standard deviation of three independent experiments (*n* = 3). * *p* < 0.05 and ** *p* < 0.01 represent significant differences compared with the control.

**Table 1 ijms-20-02023-t001:** The siRNA sequences for *TET1* knockdown.

hTET1 siRNA	5′–3′ Sense	3′–5′ Antisense
#1 siRNA	CAGGAAGUUUCUGAUACCACCUCUU	AAGAGGUGGUAUCAGAAACUUCCUG
#2 siRNA	GGCUACACGAUUAGCUCCAAUUUAU	AUAAAUUGGAGCUAAUCGUGUAGCC
#3 siRNA	GGAAGCACUGUGGUUUGUACCUUAA	UUAAGGUACAAACCACAGUGCUUCC

**Table 2 ijms-20-02023-t002:** Table of primers.

Primer Name	5′–3′ Forward	5′–3′ Reverse
*TET1*	GGCCCATATTATACACACCTTGG	GGACACCCATGAGAGCTTTTC
*IL-6*	TGCAATAACCACCCCTGACC	AGCTGCGCAGAATGAGATGA
*TNF-α*	GCCTCTTCTCCTTCCTGATCG	TCGAGAAGATGATCTGACTGCC
*CCL2*	AGAATCACCAGCAGCAAGTG	TCCATGGAATCCTGAACCCA
*CCR7*	CATAGTCTTCCAGCTGCCCT	ACAAGAAAGGGTTGACGCAG
*GAPDH*	TCTCCTCTGACTTCAACAGCGACA	CCCTGTTGCTGTAGCCAAATTCGT

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
