# Peer review of "TET1 Knockdown Inhibits Porphyromonas gingivalis LPS/IFN-γ-Induced M1 Macrophage Polarization through the NF-κB Pathway in THP-1 Cells"

_ijms, 2019, doi:10.3390/ijms20082023_

Round 1

Reviewer 1 Report

The manuscript entitled “TET1 knockdown inhibits Pg. LPS/IFN-γ-induced M1 macrophage polarization through the NF-κB pathway in THP-1 cells” by Yanlan Huang, Cheng Tian, Qimeng Li, and Qiong Xu describes the importance of TET1 in activation of NF-κB during macrophage polarization.  This study indicates the possible role of TET1 in inflammatory response upon periodontal diseases.

There are minor improvements needed:

1. Flow cytometry histogram legends should be simplified.     

2. Fig. 1B legends are slightly unclear and PMA treatment is not indicated.

3. Line 102: For cell viability, no significant difference is shown for 0, 1 and 10 ng/ml IFN-gamma for 24 and 48h.

4. Line 41: NF-κB abbreviation is nuclear factor kappa B (not beta) and the full name of the protein is nuclear factor κ-light-chain-enhancer of activated B cells.

5. Line 261, 262: IKBKE abbreviation should be corrected.

6. Line 96: Please provide the correct magnification.

7. Line 38: Interferon --> interferon

8. Line 82: macrophaes --> macrophages

9 Line 174: Enzyme --> enzyme

10. Line 298: Sirna --> siRNA

11. Line 313: Albumin Bovine --> bovine serum albumin

12. Line 313 not 325 should include Ab abbreviation

Author Response

Dear Editor,

We would like to thank you and the reviewers for providing us with very valuable and helpful comments on our manuscript. All the authors have had a chance to revise and improve this manuscript, and our final revision has been drafted according to the comments from the editor and reviewers. All revisions are highlighted in red in the manuscript.

Reviewer comments:

Reviewer: 1

Comments to the Author

The manuscript entitled “TET1 knockdown inhibits Pg. LPS/IFN-γ-induced M1 macrophage polarization through the NF-κB pathway in THP-1 cells” by Yanlan Huang, Cheng Tian, Qimeng Li, and Qiong Xu describes the importance of TET1 in activation of NF-κB during macrophage polarization.  This study indicates the possible role of TET1 in inflammatory response upon periodontal diseases.

Comments:

1. Flow cytometry histogram legends should be simplified.     

We thank the reviewer for the suggestion. We have carefully corrected the flow cytometry figures and figure legends in the manuscript. The revisions are highlighted in red in the manuscript.

2. Fig. 1B legends are slightly unclear and PMA treatment is not indicated.

We have revised the Fig. 1B legends. 100 ng/ml PMA-treated THP-1 cells were defined as M0 macrophages. The revisions are highlighted in red in the manuscript.

3. Line 102: For cell viability, no significant difference is shown for 0, 1 and 10 ng/ml IFN-gamma for 24 and 48h.

We have corrected this sentence in the manuscript.

4. Line 41: NF-κB abbreviation is nuclear factor kappa B (not beta) and the full name of the protein is nuclear factor κ-light-chain-enhancer of activated B cells.

We are sorry for our incorrect writing. We have carefully corrected the phrases throughout the manuscript. The revisions are highlighted in red in the manuscript.

5. Line 261, 262: IKBKE abbreviation should be corrected.

We are sorry for this incorrect writing. The full name of IKBKE is inhibitor of nuclear factor kappa B kinase subunit epsilon. The revision is highlighted in red in the manuscript.

6. Line 96: Please provide the correct magnification.

The magnification has been added to the manuscript.

7. Line 38: Interferon --> interferon

We have corrected this word in the manuscript.

8. Line 82: macrophaes --> macrophages

We have corrected this word in the manuscript.

9 Line 174: Enzyme --> enzyme

We have amended this word in the manuscript.

10. Line 298: Sirna --> siRNA

We have revised this phrase in the manuscript.

11. Line 313: Albumin Bovine --> bovine serum albumin

The phrase has been revised and highlighted in red in the manuscript.

12. Line 313 not 325 should include Ab abbreviation

We thank the reviewer for the valuable suggestion. We have carefully checked and corrected the phrases in the manuscript. The revisions are highlighted in red in the manuscript. 

Reviewer 2 Report

The manuscript by Huang Y. et al. describes the role of the DNA demethylase TET1 in M1 macrophage activation in periodontal infections. The topic is original, especially for the field of the study and it is very interesting. Nevertheless the study is not acceptable for publication in the current form.

Indeed several concerns remain about the described results and should be addressed.

About the Results, they should be written in more professional and detailed style. Some sentences in 2.1 Pg. LPS/IFN-γ polarized…. were too didactic and methodological. These could be avoided or re-written in more professional way.

Pag6 line 167 and Fig.5C In the histogram plots of flow cytometry did not show that TET1 expression knockdown inhibited the accumulation (it is better to say “induction or upregulation” of HLA-DR protein, as written in the paper. Also by the graph it is clear only a slight decrease of HLA-DR expression in cells transfected with siTET1.

Pag8 lines 184-189. The description of NF-kB pathway is very confusing and too general. IkBα is a molecular inhibitor of NF-kB and its phosphorylation drives its ubiquitination and proteasome degradation. P65 is an effector member of NF-kB pathway and its phosphorylation indicates its activation. Therefore, to describe the effects of TET1 knockdown on this pathway its necessary to explain in details also the meaning in terms of activation/inhibition of the pathway.

In the discussion several ways of NF-kB modulation by methylation/demethylation were mentioned. It could be interesting to investigate some of these hypotheses ( e.g. the methylation of LXRα and PPARγ1 promoters or methylation of iNOS gene) in siTET1 cells.  

Author Response

Dear Editor,

We would like to thank you and the reviewers for providing us with very valuable and helpful comments on our manuscript. All the authors have had a chance to revise and improve this manuscript, and our final revision has been drafted according to the comments from the editor and reviewers. All revisions are highlighted in red in the manuscript.

Reviewer comments:

Reviewer 2

Comments to the Author

1. About the Results, they should be written in more professional and detailed style. Some sentences in 2.1 Pg. LPS/IFN-γ polarized…. were too didactic and methodological. These could be avoided or re-written in more professional way.

We thank the reviewer for the valuable suggestion. The relevant descriptions have been revised in the result section. The revisions are highlighted in the manuscript.

2. Pag6 line 167 and Fig.5C In the histogram plots of flow cytometry did not show that TET1 expression knockdown inhibited the accumulation (it is better to say “induction or upregulation” of HLA-DR protein, as written in the paper. Also by the graph it is clear only a slight decrease of HLA-DR expression in cells transfected with siTET1.

We thank the reviewer for the suggestion. We have revised the result in the manuscript. The revisions are highlighted in red in the manuscript.

3. Pag8 lines 184-189. The description of NF-kB pathway is very confusing and too general. IkBα is a molecular inhibitor of NF-kB and its phosphorylation drives its ubiquitination and proteasome degradation. P65 is an effector member of NF-kB pathway and its phosphorylation indicates its activation. Therefore, to describe the effects of TET1 knockdown on this pathway its necessary to explain in details also the meaning in terms of activation/inhibition of the pathway.

We thank the reviewer for the valuable suggestion.

IκB kinase (IKK) consists of two subunits including IKKα and IKKβ; its activation facilitates the phosphorylation, ubiquitination and proteasomal degradation of IκBα [1]. IkBα is a molecular inhibitor of NF-κB and its phosphorylation drives its ubiquitination and proteasome degradation [2]. As an effector member of NF-κB pathway, p65 can be liberated from IκBα and then phosphorylates and translocates into the nucleus, indicating the activation of NF-κB signaling [3]. Following Pg. LPS/IFN-γ or E. coli LPS/IFN-γ treatment, the levels of phosphorylated IKKα/β, IκBα and p65 decreased in TET1 knockdown group, indicating that TET1 knockdown inhibited the NF-kB signaling pathway. The relevant descriptions have been revised in the result section. The revisions are highlighted in the manuscript.

References:

1. van Delft, M.A.; Huitema, L.F.; Tas, S.W. The contribution of NF-kappaB signalling to immune regulation and tolerance. Eur. J. Clin. Invest. 2015, 45, 529-539.

2. Yamamoto, Y.; Gaynor, R.B. IkappaB kinases: key regulators of the NF-kappaB pathway. Trends Biochem. Sci. 2004, 29, 72-79.

3. Pires, B.; Silva, R.; Ferreira, G.M.; Abdelhay, E. NF-kappaB: Two Sides of the Same Coin. Genes (Basel). 2018, 9.

4. In the discussion several ways of NF-kB modulation by methylation/demethylation were mentioned. It could be interesting to investigate some of these hypotheses ( e.g. the methylation of LXRα and PPARγ1 promoters or methylation of iNOS gene) in siTET1 cells. 

We thank the reviewer for this comment.

A previous study showed that DNMTs promoted the expression of genes involved in inflammation, such as IL-6, TNF-α, IL-1β, NO and CCL2, in mouse macrophages via the methylation of CpG-enriched sites in the liver X receptor α (LXRα) and peroxisome proliferator-activated receptor γ1 (PPARγ1) promoters; LXRα and PPARγ1 could inhibit the ability of NF-κB to drive inflammatory gene expression [1]. Tet1 and Tet2 knockdown lead to DNA hypermethylation at the Pparγ locus and reduction in Pparγ expression during adipogenic differentiation in 3T3-L1 cells [2]. TET1 promotes a significant loss of methylation of CpG sites localized near the transcription start site of iNOS gene, leading to high expression of iNOS in human fetal bone cells [3]. In an inflammatory rat burn injury model, TET1 enhances expression of iNOS by direct demethylation of iNOS gene promoter [4]. These studies indicate that DNA methylation/demethylation epigenetically regulates inflammatory responses by several different mechanisms. More studies are necessary to further illuminate the exact mechanisms of TET1-mediated NF-κB signaling pathway activation in LPS/IFN-γ-induced M1 macrophages.

References:

1. Cao, Q.; Wang, X.; Jia, L.; Mondal, A.K.; Diallo, A.; Hawkins, G.A.; Das, S.K.; Parks, J.S.; Yu, L.; Shi, H.; Shi, H.; Xue, B. Inhibiting DNA Methylation by 5-Aza-2'-deoxycytidine ameliorates atherosclerosis through suppressing macrophage inflammation. Endocrinology. 2014, 155, 4925-4938.

2. Yoo, Y.; Park, J.H.; Weigel, C.; Liesenfeld, D.B.; Weichenhan, D.; Plass, C.; Seo, D.G.; Lindroth, A.M.; Park, Y.J. TET-mediated hydroxymethylcytosine at the Ppargamma locus is required for initiation of adipogenic differentiation. Int J Obes (Lond). 2017, 41, 652-659.

3. de Andres, M.C.; Kingham, E.; Imagawa, K.; Gonzalez, A.; Roach, H.I.; Wilson, D.I.; Oreffo, R.O. Epigenetic regulation during fetal femur development: DNA methylation matters. PLoS One. 2013, 8, e54957.

4. Gong, Z.; Yuan, Z.; Dong, Z.; Peng, Y. Glutamine with probiotics attenuates intestinal inflammation and oxidative stress in a rat burn injury model through altered iNOS gene aberrant methylation. Am. J. Transl. Res. 2017, 9, 2535-2547

Round 2

Reviewer 2 Report

The current version of manuscript was improved following suggestions and comments. In my opinion it is acceptable for publication in  IJMS.